# bsAS, an antisense long non-coding RNA, essential for correct wing development through regulation of blistered/DSRF isoform usage

**Sílvia Pérez-Lluch**[1]* *, **Cecilia C. Klein**[1,2]☯, **Alessandra Breschi**[1]☯, **Marina Ruiz-Romero**[1], **Amaya Abad**[1], **Emilio Palumbo**[1], **Lyazzat Bekish**[1], **Carme Arnan**[1], **Roderic Guigó**[1,3]*

**1** Centre for Genomic Regulation (CRG), The Barcelona Institute for Science and Technology, Barcelona (BIST), Catalonia, Spain, **2** Departament de Genètica, Microbiologia i Estadística, Facultat de Biologia and Institut de Biomedicina (IBUB), Universitat de Barcelona, Barcelona, Catalonia, Spain, **3** Universitat Pompeu Fabra (UPF), Barcelona, Catalonia, Spain

☯ These authors contributed equally to this work.
* silvia.perez@crg.cat (SPL); roderic.guigo@crg.cat (RG)

**Data Availability Statement:** The authors confirm that all data underlying the findings are fully available without restriction. RNA-Seq raw data and

## Abstract

Natural Antisense Transcripts (NATs) are long non-coding RNAs (lncRNAs) that overlap coding genes in the opposite strand. NATs roles have been related to gene regulation through different mechanisms, including post-transcriptional RNA processing. With the aim to identify NATs with potential regulatory function during fly development, we generated RNA-Seq data in *Drosophila* developing tissues and found *bsAS*, one of the most highly expressed lncRNAs in the fly wing. *bsAS* is antisense to *bs/DSRF*, a gene involved in wing development and neural processes. *bsAS* plays a crucial role in the tissue specific regulation of the expression of the *bs*/DSRF isoforms. This regulation is essential for the correct determination of cell fate during *Drosophila* development, as *bsAS* knockouts show highly aberrant phenotypes. Regulation of *bs* isoform usage by *bsAS* is mediated by specific physical interactions between the promoters of these two genes, which suggests a regulatory mechanism involving the collision of RNA polymerases transcribing in opposite directions. Evolutionary analysis suggests that *bsAS* NAT emerged simultaneously to the long-short isoform structure of *bs*, preceding the emergence of wings in insects.

## Author summary

Long non-coding RNAs (lncRNAs) are transcribed regions of the genome that, unlike coding genes, are not translated to proteins. Mostly undetected until the recent development of advanced sequencing methods to profile the RNA content (the transcriptome) of the cells, we know now that in many species they rival in number protein coding genes. In the fly genome, there are more than two thousand lncRNAs, most of which of unknown function. Here we characterize the function of *bsAS*, one of the most abundant lncRNAs

processed files were deposited in the ArrayExpress database at EMBL-EBI (www.ebi.ac.uk/arrayexpress) under accession number E-MTAB-7653.

**Funding:** This work was supported by the European Community under the FP7 program (ERC-2011-AdG-294653-RNA-MAPS to R.G.), by the Spanish Ministry of Economy and Competitiveness (MEC) (BIO2011-26205 to R.G) by the Centro de Excelencia Severo Ochoa, from the CERCA Programme (Generalitat de Catalunya), and from the Spanish Ministry of Science and Innovation to the EMBL partnership. The funders had no role in study design, data collection and analysis, decision to publish, or preparation of the manuscript.

**Competing interests:** The authors have declared that no competing interests exist.

encoded in this genome. We show that *bsAS* is essential for the correct development of the fly wings, since these become very aberrant and ill-formed in mutants in which we delete this gene. We also show that the function of *bsAS* in the development of fly wings is mediated by the regulation of the expression of the *blistered* gene, a protein coding gene that overlaps in the fly genome with *bsAS*.

## Introduction

In recent years, the annotation of long non-coding RNAs (lncRNAs) has expanded substantially thanks to the spread of high-throughput RNA sequencing technologies [1]. Natural Antisense Transcripts (NATs) are most commonly defined as fully processed lncRNAs which overlap protein coding genes on the opposite strand with or without exonic complementarity [1]. Several roles in genomic regulation (in *cis* or in *trans*) have been reported for NATs in metazoans [2,3], including gene expression regulation of the host protein coding gene, DNA methylation, chromatin modifications and RNA editing. NATs have been related to different human diseases, such as cancer, parkinsonism, alzheimer or autism and they have become even more relevant since they emerged as putative targets for gene therapy (revised in [4]). NATs have also been related to regulation of alternative splicing in vertebrates [5]. In a well understood case, a transcript antisense to *Zeb2* promotes intron retention of the sense gene and affects its translation in human epithelial-mesenchymal transition [6].

The fruit fly genome is estimated to encode more than two thousand lncRNAs [7], of which about one third are NATs. However, only few have a characterized function. For instance, *roX* is a well-studied antisense transcript involved in chromatin regulation and required for dosage compensation of the male X chromosome [8]. Or the *male specific abdominal* (*msa*), a lncRNA expressed specifically in fly male abdomen, has been related to fertility in flies [9]. Here, we investigate the role of NATs in the regulation of alternative splicing during *Drosophila melanogaster* development. We have specifically identified a lncRNA (*CR44811*), the expression of which regulates the isoform usage of the sense transcription factor *blistered/Drosophila Serum Response Factor* (*bs/DSRF*, or simply *bs* gene), a gene essential for the proper development of wings. To form the final wing during fly development, the wing disc undergoes a process of evagination and ultimately develops a characteristic patterning of veins and interveins (reviewed in [10]). Veins are epithelial formations that carry the trachea and nerves of the adult wing [11] and their specification is tightly regulated during the fly development. Vein and intervein regions are specified from the third instar larvae through the action of *bs/DSRF*, which has been reported to be expressed specifically in intervein regions [10]. *Blistered* mutants show dramatic defects in the wings, such as reduction of the intervein tissue, and overgrowth and appearance of extra veins [12–14]. A part from its role in wing morphogenesis, *bs* has also been related to neural processes such as sleep, visual orientation and working memory in *Drosophila* [15,16]. In mammals, the homologue *SRF* gene has been involved in muscle development [17] and neuronal plasticity [18], among other processes.

The protein encoded by *bs*, DSRF, is, in fact, present in two different isoforms in the fly, which differ in length. We found that the long isoform of *bs* is preferentially expressed in the eye, while the short isoform is preferentially expressed in the wing, particularly in intervein regions. We show that the usage of these isoforms is regulated in a tissue-specific manner by the expression of the *bs* NAT *CR44811* (herein *bsAS*). The transcription of *bsAS*, which occurs specifically in intervein regions, impairs the expression of the long isoform of *bs*, promoting, in consequence, the relative expression of the short isoform. The overexpression of the long

isoform in *bsAS* mutants induces the formation of extra vein tissue in adult wings. Our results also suggest that the differential regulation of *bs* expression in different tissues could be driven by the usage of tissue-specific promoters. In particular, the *bs* promoter specifically active in wing intervein regions appears to interact, through the formation of a genomic loop, with the *bsAS* promoter. Phylogenetic analysis shows that the emergence of *bsAS* within arthropoda is contemporary to the functional distinction between short and long *bs* isoforms and precedes the appearance of the wings in insects.

## Material and methods

### *Drosophila* strains

Fly strains used for this work are: wild-type (CantonS), *nub-GAL4; UAS-GFP* and *sal^{E/Pv}-GAL4* (kindly gifted by Corominas' lab, UB), *w^{1118}* (kindly gifted by Gebauer's lab, CRG), *bs-GAL4* (B67082), *rn-GAL4/TM3* (B7405), *UAS-RNAi_bs_long* (B26755) and *bsAS -/-*, *UAS-bsAS*, *UAS-bsIsoB* and *UAS-bsLongIsos* (generated in the lab). Flies were grown in standard media at 25˚C.

### *In situ* hybridization and immunohistochemistry

*In situ* hybridizations and immunostaining were carried out according to standard protocols. For *in situ* hybridization, *bsAS* DNA probe was synthesized by conventional PCR using a PCR DIG Labeling Mix (Roche). A biotinylated sense primer was used to bind the PCR product to streptavidin beads. Anti-sense probe was purified by denaturalization of the DNA attached to the beads. Primers used for probe synthesis are listed in S4 Table. Peroxidase conjugated anti-digoxigenin and Tyramide signal amplification (TSA, Life Technologies) were used for fluorescent *in situ* hybridization (FISH). DSRF immunostaining was performed with mouse anti-DSRF (Active Motif, 1:200). Fluorescently labeled secondary antibody anti-mouse Alexa 594 was from Life Technologies. Discs were mounted in SlowFade (Life Technologies) supplemented with 1 μM DAPI (Life Technologies) to label nuclei. Third instar larvae wing and eye discs were analyzed with a SPE confocal microscope (Leica) at the Advanced Light Microscopy Unit of the Centre for Genomic Regulation (CRG, Barcelona, Spain). For all *FISH* and immunostainings from 6 to 12 imaginal discs were analyzed. Quantification of DSRF staining from vein and intervein regions was performed with ImageJ software. Intervein region between veins 3 and 4, and vein 3 were selected for quantification and average fluorescence intensity was obtained. For discs overexpressing *bs* long isoform under the control of *rotund* driver, a comparable region was selected.

### Chromatin immunoprecipitation

As starting material, 100 wt and *bsAS-/-* third instar larvae wing discs were used per experiment. Imaginal discs were manually dissected and pooled in 1 mL PBS-0.01% TritonX100. Formaldehyde was added to a final concentration of 1% and tissues were fixed for 10 minutes at room temperature in a rotating wheel. Sonication was performed in a Diagenode Bioruptor for 15 minutes at high intensity with ON/OFF alternate pulses of 30 seconds. Sheared chromatin was aliquoted and flash frozen in liquid nitrogen. Chromatin immunoprecipitation assays were next performed as previously described [19]. For immunoprecipitation, 2 μg of anti-H3K4me3 (Abcam/ab8580) were used. Enrichments were analyzed by Real-Time PCR. Primers used for ChIP enrichment detection are listed in S4 Table. Three independent biological replicates were performed per experiment.

## RNA isolation, library preparation and sequencing

RNA from 20 to 40 imaginal discs was extracted with ZR-RNA MicroPrep Kit from Zymo Research following the manufacturer's instructions. Sequencing libraries were prepared using TruSeq Stranded mRNA Library Preparation Kit from Illumina and following the manufacturer's instructions. Sequencing was performed in a HiSeq sequencer from Illumina at the Ultrasequencing Unit of the Centre for Genomic Regulation (CRG, Barcelona, Spain). A minimum of 50 million paired-end 75 bp-long reads were obtained per replicate and two replicates were performed per each tissue.

## Cell isolation from wing imaginal discs

From 150 to 200 third instar larvae wing imaginal discs from animals expressing the GFP under the *bs* driver were dissected and disaggregated in 10× trypsin solution (Sigma, T4174) for 1 hour at room temperature, as previously described [20]. To discard dead cells, DAPI was added to the samples at a 1 µg/ml final concentration. Cells were sorted according to the GFP intensity on an Influx sorter (BD), in the Flow Cytometry Unit, from the Centre for Genomic Regulation (CRG) and University Pompeu Fabra (UPF, Barcelona, Spain). GFP positive and negative cells were isolated and RNA was extracted as above.

## Retro-transcription and Real-Time PCR analyses

Retro-transcriptions and qPCRs were performed as described previously [19]. Primers used for Real-Time PCR are listed in S4 Table. From two to five independent biological replicates were performed per experiment.

## CRISPR-Cas9-induced deletion in flies

Guide RNAs around the *bsAS* TSS were designed using the CRISPR Target Finder from the flyCRISPR portal (http://flycrispr.molbio.wisc.edu/). Sequences of the gRNAs are listed in S4 Table. gRNAs were cloned into a *BbsI* digested pU6-*BbsI*-chiRNA vector [20] following the protocol in flyCRISPR portal. A mixture of pU6-*BbsI*-gRNA1 and pU6-*BbsI*-gRNA2 was injected into embryos expressing Cas9 protein under the control of the *vasa* driver. Injection was performed in Rainbow Transgenic Flies Inc. (Camarillo, USA). Injected flies (F0) were crossed in groups of 4 males and females. F1 flies were allowed to mate and ley eggs for 10 days before screening. Sequential crosses were performed until the flies presenting the deletion were identified and isolated. Mutant flies were crossed with flies carrying a CyO balancer and homozygous mutants were isolated (S2A Fig). Flies were screened by conventional PCR. Briefly, genomic DNA was extracted from groups of 4 flies by smashing the animals in lysis buffer containing 0.5% NP40, 10 mM TrisHCl pH8.0, NaCl 150 mM, EDTA 2 mM and proteinase K 1 mg/mL. Genomic DNAs were incubated for 2 h at 50˚C and centrifuged at top speed for 5 minutes to remove the remaining fly fragments. 2 µL of the extracts were used for each PCR. Primers used for the screening are listed in S4 Table.

## Transgenic constructs

For transgenesis, *bs* isoform B and long isoform (common to isoforms A and C) cDNAs were amplified by rtPCR (primers used for the amplifications are listed in S4 Table) and inserted by Gibson cloning into a pUAST-AttB vector digested with *EcoRI*. Transgenes were integrated using the phi31 integrase system in fly strain J36, in the chromosome III. Injection for transgenesis was performed in the *Drosophila* Injection Service from the Institut de Recerca Biomèdica (IRB, Barcelona, Spain).

## Chromosome Conformation Capture (3C)

Chromosome conformation capture protocol was adapted from previous reports [21]. For each experiment a 3C library and a genomic library were performed. For genomic libraries, genomic DNA from 5 third instar larvae was extracted by smashing the animals in lysis buffer containing 0.5% NP40, 10 mM TrisHCl pH8.0, NaCL 150 mM, EDTA 2 mM and proteinase K 1 mg/mL. Genomic DNA was incubated for 2 h at 50˚C, centrifuged at top speed for 5 minutes to remove the remaining larval fragments, purified by performing two rounds of phenol:chloroform extraction and precipitated with ethanol. For 3C libraries, 150 wing imaginal discs were manually dissected and pooled in 1 mL PBS-0.01% TritonX100. Formaldehyde was added to a final concentration of 1% and tissues were fixed for 10 minutes at room temperature in a rotating wheel. Discs were washed in 1 mL of PBS-0.01% TritonX100-0.125 M glycine for 5 minutes at room temperature and finally resuspended in 500 μL of lysis buffer (10 mM Tris pH 8, 10 mM NaCl, 0.2% NP40 1x Complete Protease Inhibitor Cocktail) and flash frozen in liquid nitrogen. To perform the libraries, 5 μg of genomic DNA and the fixed 150 wing discs were equilibrated in 164 μL of digestion mix (20 μL *HhaI* 10x buffer, 5 μL 10% SDS and 139 μL H$_2$O) for 1 h at 37˚ in a ThermoMixer. 32 μL of 10% TritonX100 were added to each library and incubated for 1 h at 37˚C. Digestion was performed by adding 2 μL of *HhaI* restriction enzyme (New England Biolabs) and incubating for 2 h at 37˚C. After first digestion, 2 μL of restriction enzyme were added and samples were incubated over night at 37˚C. Digestions were inactivated for 20 minutes at 65˚C and cooled on ice. To perform the ligation, 165 μL of ligation mix (2x T4 ligation buffer + 4 μL T4 ligase) were added to the libraries. Ligations were incubated for 4 h at 18˚C. Libraries were de-crosslinked by adding 2 μL of proteinase K 20 mg/mL and incubated over night at 65˚C. Remaining RNAs in the samples were removed by RNase treatment for 30 minutes at 37˚C. Libraries were finally purified by phenol:chloroform extraction and precipitated in ethanol. Quantification of interactions was performed by Real-Time PCR, comparing the interaction observed in the regions of interest to a known interacting region (positive control). Primers used for Real-Time PCRs are listed in S4 Table. At least, three independent biological replicates were performed per experiment.

## RNA-Seq processing

Data was processed using grape-nf (available at https://github.com/guigolab/grape-nf). RNA-Seq reads were aligned to the fly genome assembly dm6 [22] using STAR 2.4.0 software [23] with up to 4 mismatches per paired alignment using the FlyBase genome annotation r6.05 [7]. Only alignments for reads mapping to ten or fewer loci were reported. Gene and transcripts TPMs were quantified using RSEM [24]. Tracks were visualized as a Track Hub at the UCSC Genome Browser [25].

## NATs in the fly genome and in larval tissues

Antisense lncRNAs were inferred from their overlap and orientation relative to protein coding genes. A minimum overlap of 1 exonic nucleotide was required to classify the pairs into the following configurations: (i) 5' head-to-head: first exon overlap; (ii) 3' tail-to-tail: last exon overlap; (iii) internal: lncRNA is within the protein coding gene; (iv) external: protein coding gene is within the lncRNA; and (iv) complex: overlap of more than 2 genes.

Pairs of ncRNAs antisense to mRNAs with more than 1 isoform and expressed at least 0.5 TPMs in at least both replicates of any L3 sample were selected for further analyses. Ratio of exon coverage over the gene coverage was computed using bwtool summary [26]. Candidate pairs were ranked based on the Pearson's coefficient of correlation of exon ratio and antisense

expression and on the standard deviation of exon ratios. Exons showing a standard deviation higher than 0.1 and an absolute correlation higher than 0.7 were selected.

## Differential gene expression analysis of mutant versus wild type samples

Pairwise differential gene expression (DEG) analysis between wild type and mutant samples was performed using EdgeR [27]. Only genes expressed at least 5 TPM in at least two samples were selected for this analysis. DEG selected showed $\log_2$ fold change $> 2$ and FDR $< 0.01$. Biological Processes from Gene Ontology database enriched in upregulated and downregulated datasets were assessed by using GOstats [28] and visualized with ReviGO [29], with a p value cutoff of 0.001.

## Promoter and ChIP-Seq analysis

To search for transcription factor binding sites (TFBSs) in *bs* and *bsAS* TSSs we used the Promo 3.0 tool [30]. We selected TFBSs with a matrix dissimilarity rate less or equal than 5% that were present both in *bs* TSS1 and *bsAS* TSS but not in *bs* TSS2. Available ChIP-Seq data of GAF [31], RNA PolII [32], Pc and Ph [33] were aligned to the fly genome (dm6) using GEM-Mapper [34] with up to 2 mismatches per read using the dm6 genome assembly [7,22]. ChIP-Seq data from modENCODE TFs were visualized at UCSC Genome Browser from the ENCODE Portal [35].

## Evolutionary conservation

To track down the evolutionary conservation of *bs* and *bsAS*, we first analyzed RNA-Seq data of whole body and head of female *Drosophila* species from modENCODE (GSE44612 [36]). Next we investigated the expression of *bs* and *bsAS* in stranded RNA-Seq data of *Apis mellifera* (whole body: GSE83437 [37] and head: GSE87001 [38]; UCSC apiMel2 assembly; modelRefGene genome annotation available at UCSC), in stranded RNA-Seq data of *Anoplophora glabripennis* (PRJNA274806;) in unstranded RNA-Seq data of *Folsomia candida* (PRJNA239929; RefSeq assembly accession: GCF_002217175.1; NCBI *Folsomia candida* Annotation Release 100) and in stranded RNA-Seq of *Daphnia pulex* (GSE103939; assembly GCA_000187875.1; annotation ENSEMBL release-26). Finally, we manually annotated *bs* isoforms across metazoans taking advantage of available RNA-Seq data and relying on split reads to properly annotate the exon junctions, which were visualized with ggsashimi [39]: *Anopheles gambiae* (GSE55453 [40]; GSE59773 [41]), *Apis mellifera* (GSE52289 [42]; GSE65659 [43]; SRP068487), *Daphnia pulex* (DRP002580), *Strigamia maritima* (SRP041623), *Tetranychus urticae* (GSE31527 [44,45]), *Crassostrea gigas* (GSE31012 [46]; SRP058882), *Gallus gallus* (GSE41338 [47]), and *Nematostella vectensis* (GSE46488 [48]). *Homo sapiens bs* annotation was directly extracted from GENCODE v19 [49]. Data was processed using grape-nf (available at https://github.com/guigolab/grape-nf) with the same parameters as our RNA-Seq data.

# Results

## Natural antisense transcription in *Drosophila melanogaster*

The *D. melanogaster* genome encodes 16,698 genes, including 13,920 protein coding genes, 2,433 lncRNAs and 308 pseudogenes (FlyBase v6.05 [7]). Although antisense lncRNAs are not explicitly annotated, they can be inferred from their overlap and orientation relative to protein coding genes. We calculated that 855 lncRNAs (35%) overlap 873 protein coding genes in antisense orientation (Natural Antisense Transcripts, NATs), forming 953 sense/antisense (SA) pairs, a number in line with previous reports [50,51]. The coding (sense) and non-coding

(antisense) gene pairs, can be arranged in different configurations (Fig 1A): in most of the pairs, the lncRNA is fully included within the protein coding gene (391 pairs, 41%), followed by 5' head-to-head pairs (165 pairs, 17%) and 3' tail-to-tail pairs (61 pairs, 7%). Only in a minority of the pairs (28 pairs, 3%), the protein coding gene is fully included in the lncRNA, while the remaining 32% of the pairs (308) are arranged in complex configurations involving three or more genes (Fig 1A). Remarkably, fly protein coding genes involved in SA pairs are significantly associated to developmental and morphogenetic processes (Fig 1B). This strongly suggests that NATs might play a relevant regulatory role in fly development.

With the aim of identifying NATs with potential regulatory function during fly development, we performed strand-specific RNA-Seq of three imaginal tissues of *D. melanogaster* third instar larvae (L3): eye-antenna (EAL3), leg (LL3) and wing (WL3) (S1A Fig and S1 Table). Of the 953 SA pairs, we found 145 (about 15%) in which the two members of the pair were detected with more than 0.5 TPMs (Transcripts per Million Mapped Reads [24]) in at least one tissue, albeit not necessarily the same. Since splicing regulatory activity has been documented for a human NAT [6] and extensive relationship between antisense transcription and splicing has been hypothesized in human [5], we explored specifically the relationship between NAT expression and alternative transcript usage across fly larval samples. Of the 145 SA pairs with both genes expressed in at least one tissue in L3, 104 pairs (72%, 102 genes) involve a protein coding gene with multiple isoforms. We compared the expression of these NATs with the inclusion of the 964 exons of the sense protein coding genes, focusing on the exons the inclusion of which changed the most (see Material and Methods). We found a few cases of relationship between expression of the NAT and the splicing of the sense protein coding genes—the most notable being the case of the *bs/DSRF* gene (S1B Fig).

The *bs/DSRF* gene encodes for two protein isoforms, both carrying the DNA binding domain MADS-box, but differing at the terminal end, one much longer than the other (449 vs. 355 amino acids). The long isoform is encoded by two transcripts (variants A and C in Fig 1C), that use two different transcription start sites (TSS2 and TSS1, respectively), and differ only in the first exon. The other protein isoform is encoded by a shorter two-exon transcript (variant B), which shares TSS1 with isoform C. The two long isoforms contain a 26 Kb intron, which holds the NAT *CR44811* (*bsAS* from now onwards due to its overlap with the *bs* gene), and a coding gene of unknown function (*CG44812*).

The RNA-Seq data reveals, a contrasting pattern of *bs* isoform expression between wing and eye and leg tissues (Fig 1C and 1D and S2 Table). First, the long isoforms of *bs* are globally expressed at similar levels in eye, leg and wing imaginal discs. However, in the eye, only isoform A is expressed, while wing mostly expresses isoform C (Fig 1D). Second, the short isoform B is only expressed in the wing, and at much higher levels than the long isoforms. Therefore, *bs* is globally much more expressed in the wing than in the eye and the leg. Third, the NAT *bsAS* also shows contrasting expression between these tissues, being very highly expressed in wing—where it is one of the most highly expressed lncRNAs in the fly genome (S1C Fig) —but very low in eye and leg imaginal discs. We independently confirmed the expression of the *bs* isoforms and of *bsAS* by qPCR in wing and eye at third instar larvae and late pupa stages (Fig 1E). The restricted expression of *bsAS* in wing imaginal discs is also supported by the presence of trimethylation of lysine 4 of histone H3 (H3K4me3, mark associated to active promoters) in wing, but not in eye imaginal discs (S1D and S1E Fig). In situ hybridization in wing imaginal discs reveals that the expression of *bsAS* is restricted to the intervein regions, like that of *bs* (Fig 1F), suggesting that these two genes may be co-regulated in wing imaginal discs. Since isoforms B and C share TSS1, and isoform A is associated to TSS2, these observations also suggest that TSS1 is mostly active in the wing, and TSS2 is mostly active in the eye.

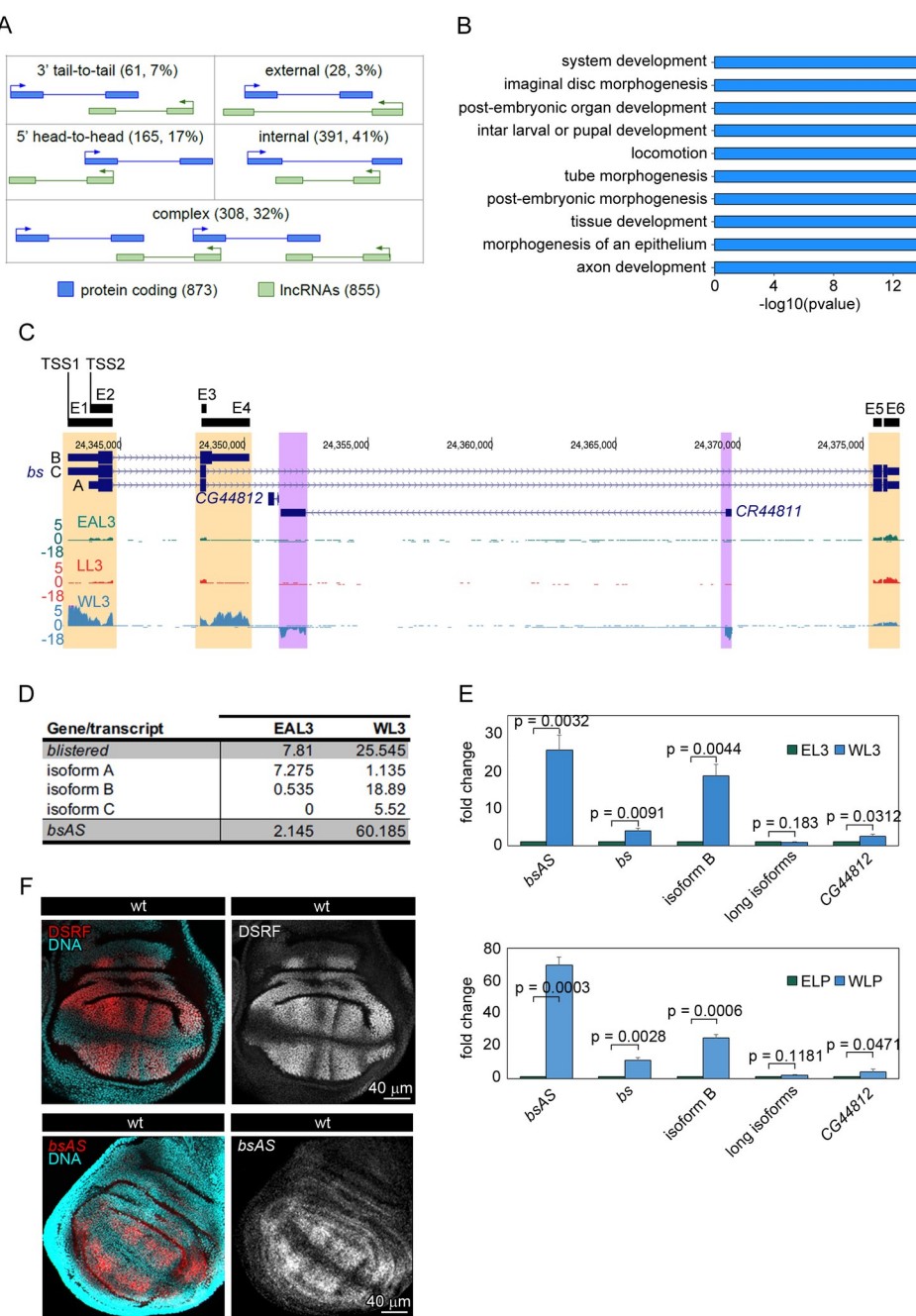

**Fig 1. Natural antisense transcription in *Drosophila melanogaster*. (A)** Configuration of sense/antisense (SA) pairs identified in *D. melanogaster* genome. We have identified 855 lncRNAs overlapping 873 coding genes in the fruit fly annotation. Around a 32% of the SA pairs present a complex configuration, meaning that one lncRNAs overlaps more than one coding gene or the other way around. **(B)** Gene Ontology Term Enrichment analysis of coding genes overlapping antisense transcripts. Sense genes are mainly related to development and morphogenesis categories. **(C)** Expression of the *bs* locus in the different tissues. Isoform B of *bs* is expressed mainly in wings, correlating with the highest expression of the antisense lncRNA *CR44811*. In the other tissues, the isoform A seems to be the predominant one. In the leg, also some expression of the lncRNA is observed. Regions showing differences in expression between tissues are highlighted (*bs* in orange and *CR44811* in purple). **(D)** Expression of *bs* isoforms and *bsAS* in eye-antenna and wing imaginal discs at third instar larvae. The expression values, expressed in TPMs, represent the mean of two independent biological replicates. **(E)** Relative expression of *bs* and *bsAS* in wings and eyes at third instar larvae (L3 – upper panel–) and late pupa (LP –lower panel–) stages. *bsAS* and the short isoform B of the coding gene are much higher expressed in wings than in eyes, whereas there are not significant differences in the expression of the long isoforms. The protein coding gene *CG44812*, of unknown function, is almost not expressed in these tissues. Error bars

depict Standard Error of the Mean (SEM) of at least three biological replicates. Statistical significance was computed by one-sample t-test. **(F)** Expression pattern of DSRF and *bsAS* in wt wing imaginal discs. Upper panels, immunostaining of DSRF (red and grey) in wt WL3. DSRF is expressed in the intervein region of the wing pouch. Lower panels, *in situ* hybridization of *bsAS* (red and grey) in WL3. The expression of *bsAS* mimics DSRF expression pattern in the intervein regions. Nuclei, stained with DAPI, are depicted in blue in all cases.

## *bsAS* controls *bs* isoform usage during fly development

In summary, thus, in wing, *bsAS* is highly expressed, coinciding with the high expression of the short isoform B. In contrast, in eye, *bsAS* is not expressed, nor short isoform B. Since the expression of *bsAS* seems to be associated to the expression of short isoform B, this suggests that *bsAS* could play a role in regulating isoform usage of *bs*. To test this hypothesis, we used CRISPR/Cas9 to induce a 560 bp deletion surrounding the TSS of *bsAS* (Figs 2A and S2). This, effectively abolishes expression of this gene in L3 and LP in wing imaginal discs (Fig 2B). Knocking out *bsAS* has little effect on the expression of the short isoform B, but induces over-expression of the long isoforms. As a result, there is an overall increase in the expression of *bs* in the *bsAS* mutant wings compared to wt (Fig 2B). This increase is particularly significant at protein level, both in heterozygous and homozygous *bsAS* mutant wings, and it is observed both at intervein and vein regions (Fig 2C and 2D). Although the expression of *bsAS* in the eye imaginal disc is very low, its deletion induces the overexpression of DSRF in few determined cells posterior to the morphogenetic furrow, but this does not produce any observable pheno-type in adult animals (S3A and S3B Fig). Animals carrying the deletion exhibited, in contrast, notable defects in wing development, which are much stronger in homozygous mutant flies, and resembling those of the *bs* mutant flies [12–14] (Fig 2E, 2F and 2G). These were blistered, presenting necrotic regions and strong defects in vein/intervein patterning, in particular, extra vein tissue in intervein regions.

Ectopic expression of *bsAS* did not rescue the wing phenotype in the mutant (Figs 2H and S3C), suggesting a role for *bsAS in cis*, likely due to the transcription process rather than to the *bsAS* transcript itself. We further observed that the extra vein phenotype is a consequence of the increase in expression of the long isoforms of *bs* in the *bsAS* mutant, since similar wing defects were observed when overexpressing the long isoform of *bs* in the wing pouch, even though the levels of the overexpressed protein were lower than in the heterozygous *bsAS* mutant (Figs 2I, S3D and S3E). The overexpression of an interference RNA targeting the long isoforms of *bs* was able to rescue the wild type vein patterning in a heterozygous *bsAS* mutant background and partially rescue the *bsAS* homozygous phenotype (Figs 2J and S3F), further confirming the role of the long isoform of *bs* in promoting vein differentiation. Overexpres-sion of the short isoform B, on the other hand, induces strong defects in wing development and even animal death (S3G and S3H Fig), suggesting that the short DSRF protein is toxic when expressed in excess or in tissues where it is not normally expressed.

## *bsAS* deletion induces overexpression of neural genes

To monitor the molecular changes underlying the *bsAS* mutant phenotype, we performed RNA-Seq of mutant wing imaginal discs in L3 and LP (Fig 3A and S2 Table). RNA-Seq con-firmed the targeted expression results in Fig 2B, and revealed that the long isoform preferen-tially induced in the *bsAS* mutant is isoform C, while the expression of isoform A is less affected. We observed few genes differentially expressed comparing the mutant with the wt flies in L3 (27 up and 29 down regulated), but many more in LP: 275 upregulated and 260 downregulated genes (S3 Table). There were not clear functional categories enriched among downregulated genes (Fig 3B) but, after closer inspection, we identified many genes involved

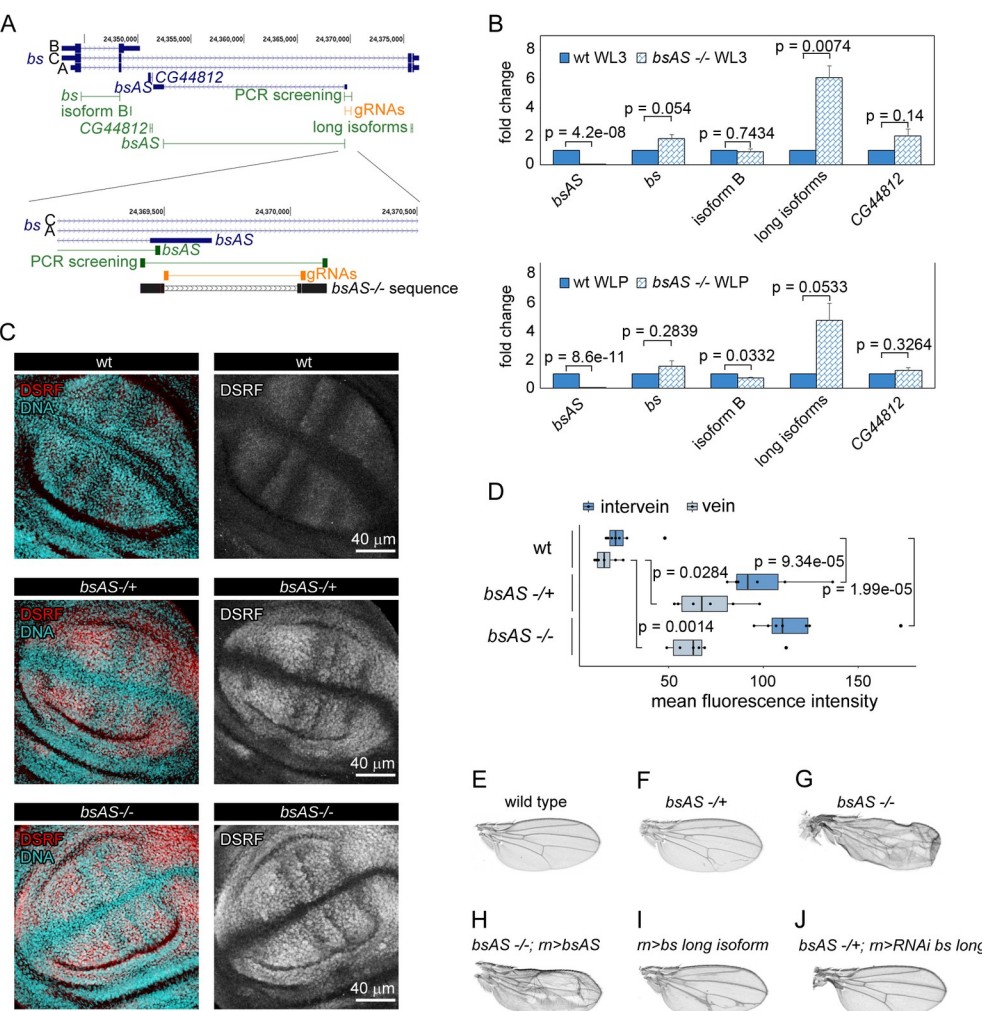

**Fig 2. *bsAS* controls *bs* isoform usage during fly development. (A)** Distribution of primers and CRISPR gRNAs along the *bs* locus. Upper panel, primers amplifying the different isoforms of *bs* and *bsAS* are depicted in green; gRNAs used to induce the deletion of *bsAS* are depicted in orange. Lower panel, zoom of the lncRNA TSS region. Sanger sequence of *bsAS* mutant flies is depicted in black. A 560 bp long deletion was induced around the TSS of *bsAS*. A small deletion of 2 nucleotides at the 5' region of the TSS is also observed. **(B)** Relative expression of *bs* and *bsAS* in *bsAS* -/- and wt L3 (upper panel) and LP (lower panel) wings. The lncRNA is not expressed in *bsAS* -/- wings. The overall expression of the *bs* gene seems to be slightly higher in mutant than in wt wings, especially in L3 stage; however, when comparing the expression of *bs* isoforms, we do not observe differences in the expression of the short one between the mutant and the wt, while the long isoforms are overexpressed up to 6-fold in the mutants. Error bars depict Standard Error of the Mean (SEM) of at least three biological replicates. Statistical significance was computed by one-sample t-test. **(C)** Expression pattern of DSRF in *bsAS* mutant wing imaginal discs. Immunostaining of DSRF (red and grey) in wt (upper panels), *bsAS* -/+ (middle panels) and *bsAS* -/- (lower panels) WL3. Nuclei, stained with DAPI, are depicted in blue in all cases. **(D)** Quantification of DSRF staining in vein and intervein regions of wt and *bsAS* mutant wing imaginal discs. Both vein and intervein regions exhibit a significant increase of DSRF staining in *bsAS* mutants compare to wt wings. Statistical significance was computed by two-sample t-test. **(E-J)** Adult wings from males. **(E)** Wild type. **(F)** *bsAS* heterozygous mutant. The wings present extra veins invading intervein tissue. **(G)** *bsAS* homozygous mutant. The wings are creased and vein pattern is completely impaired. **(H)** *bsAS* homozygous mutant overexpressing transgenic *bsAS* under the control of *rotund* (*rn*) driver. The mutant phenotype is not rescued by the overexpression of the lncRNA. **(I)** Wings overexpressing *bs* isoform A under the control of the *rn* driver. Extra veins appear within intervein tissue, resembling the phenotype of *bsAS* heterozygous mutants. **(J)** *bsAS* heterozygous mutant wings overexpressing the RNAi against *bs* long isoform. The overexpression of the RNAi rescues the extra vein phenotype in *bsAS* heterozygous mutant background.

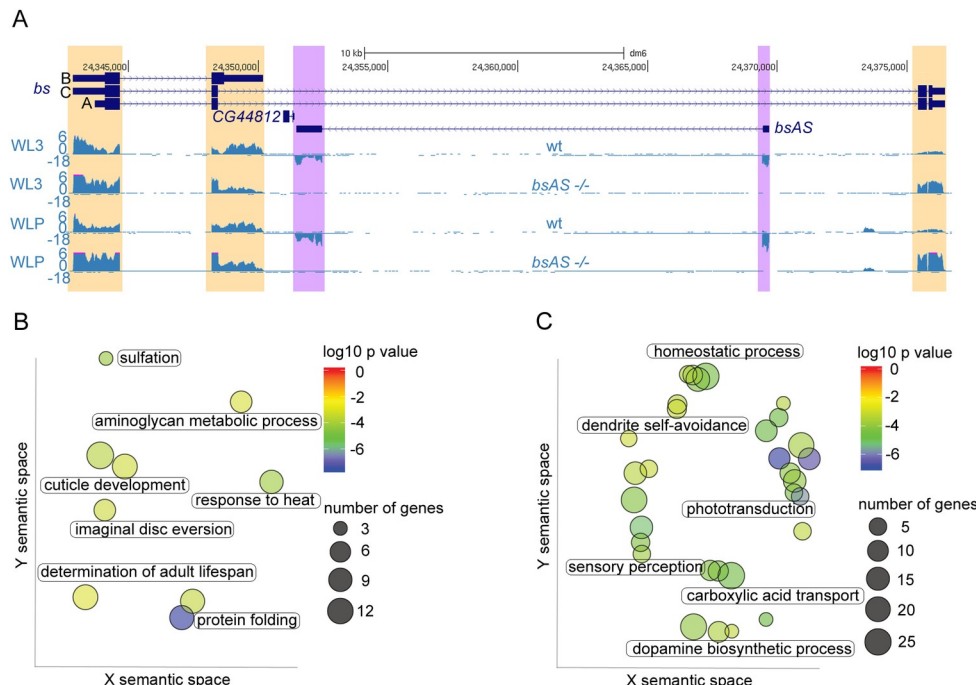

**Fig 3.** ***bsAS* mutation induces overexpression of neural genes in wing imaginal discs. (A)** Expression of *bs* locus in wt and *bsAS* -/- wings at L3 (upper panels) and LP (lower panels) –note that wt tracks coincide with tracks in Fig 1C–. Differences in expression between wt and mutant tissues are highlighted in purple for the lncRNA and in orange for the coding gene. The depletion of *bsAS* expression after *bsAS* TSS deletion is confirmed in the RNA-Seq experiments, as well as the consequent overexpression of the long *bs* isoform C. **(B)** Gene Ontology Term Enrichment analysis of downregulated genes in *bsAS*-/- WLPs. Few GO categories were enriched among downregulated genes in *bsAS*-/- LP wings, among them, cuticle development and imaginal disc eversion. **(C)** Gene Ontology Term Enrichment analysis of upregulated genes in *bsAS*-/- WLPs. Upregulated genes are enriched in categories mainly related to neural fates, such as phototransduction, sensory perception and dopamine biosynthesis. In **(B)** and **(C)**, each spot represents an enriched category and labels are representative of clusters of categories; color of the spots represents the p value of enrichment; size of the spots represents number of genes belonging to each category. Axes represent semantic space from ReviGO [29].

in wing morphogenesis and cell adhesion. These include *Sox102F*, that regulates the expression of the *Wnt* pathway and has been related to wing vein development and patterning [52], *ImpE2* and *ImpE3* that are ecdysone-inducible genes related to imaginal disc eversion [53–55] and *multiple edematous wings* (*mew*) and *miniature* (*m*), that are genes related to cell adhesion and wing morphogenesis [56,57]. In contrast, upregulated genes were strongly enriched in functions related to photo transduction, dopamine biosynthesis and other neural specific functions (Fig 3C). Among the genes upregulated in mutant wings we found *Dscam1* and *Dscam4*, cell adhesion molecules related to axon guidance and neural development [58,59], and *inaC*, *inaD* and *ninaE*, genes involved in the detection and response to light stimuli (for a review see [60]).

All together, these results lead us to hypothesize that the expression of the *bs* long isoforms may induce the expression of neural specific genes, and that *bsAS* prevents the development of veins in the wing intervein regions by both impairing the expression of the long isoform C, and promoting the expression of short isoform B. To assess this hypothesis, we analyzed the expression of the *bs* locus in vein and intervein regions (S4 Fig). Wing imaginal discs expressing the GFP under the control of the *bs* driver were disaggregated and GFP positive and negative cells were isolated by flow cytometry. Expression analyses indicate that, indeed, *bsAS* and *bs* isoform B are mainly expressed in the intervein region (GFP positive cells), while the long

isoforms of *bs* are expressed at similar levels in veins and in the rest of the imaginal disc, suggesting that they are involved in additional roles a part from intervein differentiation.

## *bsAS* and *bs* are co-regulated through the interaction of their TSSs

Since *bs* isoforms B and C share TSS1, we investigated whether physical interactions between the TSS of *bsAS* and TSS1 could underlay *bsAS* regulation of *bs* isoform usage. Using publicly available HiC data in Kc167 cells [61], we did find an enrichment in contacts between the TSS of *bsAS* and TSS1, even though *bs* and *bsAS* are poorly expressed in these cells [62] (Figs 4A and S5A). No other contacts were identified in further regions around the *bs* locus (S5B Fig). We next performed 3C assays in L3 wing and eye imaginal discs, interrogating the interaction of *bsAS* TSS with ten different regions along the *bs* locus (including TSS1 and TSS2) (Fig 4B). We detected a significant interaction with TSS1 both in wing and eye, but not with TSS2, although it is very close to TSS1 (Fig 4C). This interaction seems to be very stable, present even in tissues in which TSS1 and *bsAS* are inactive, such as Kc197 cells and eye tissue. We detected a second interaction with a region known to be a hot spot for transcription factor binding in fly embryos [63] (S6A Fig). Interactions between TSS1 and a region adjacent to *bsAS* TSS (bait2), which were maintained in wt wings and eyes, were dramatically impaired in *bsAS* mutants (Fig 4D), indicating that sequences within this region are likely responsible for the interaction with TSS1.

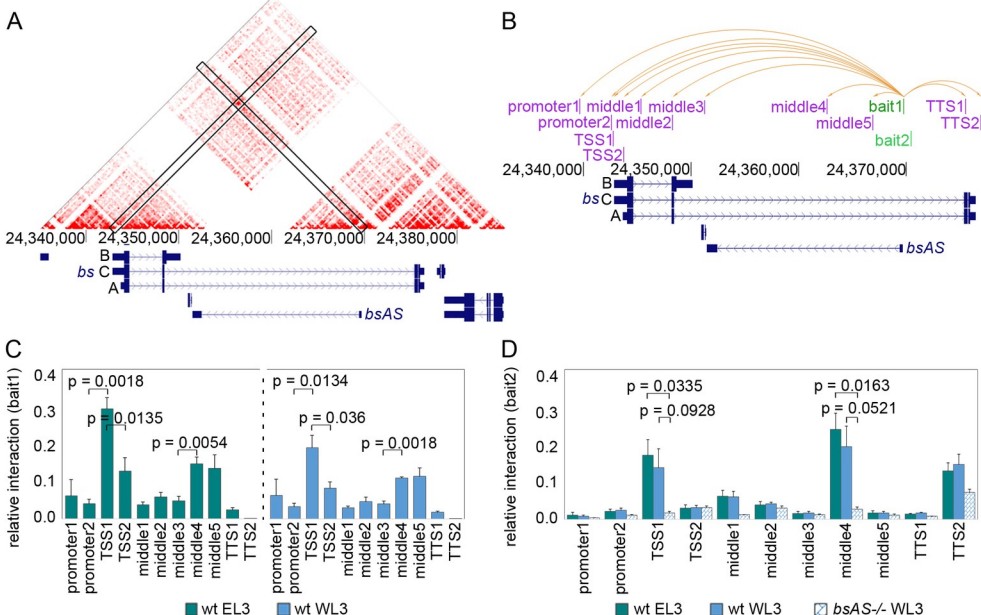

**Fig 4. *bsAS* and *bs* are co-regulated through the interaction of their TSSs. (A)** JuiceBox tool [76] was used to visualize high-resolution HiC maps generated in Kc167 cells [61]. A strong interaction signal is observed in the intersection between *bs* TSS1 and *bsAS* TSS. **(B)** Representation of *bs* locus and the tested interactions inside this region. Two different baits were anchored at *bsAS* TSS (in green) and interaction enrichment was assessed for each bait by 3C experiments against all depicted regions (in pink). Orange arrows represent all tested interactions for bait1. The same interactions were tested for bait2. **(C)** Interaction between bait1 and *bs* locus in wild type eyes (left panel) and wings (right panel). Interaction enrichment is significantly higher between *bsAS* TSS and TSS1 than between the *bsAS* TSS and the other tested regions, both in wings and eyes. A significant increase in relative interaction is also detected between *bsAS* TSS and middle regions 4 and 5. **(D)** Interaction between bait2 and *bs* locus in wild type wings and eyes and *bsAS*-/- wings. The interaction between bait2 and TSS1 is still enriched in wild type tissues, but it is heavily impaired in *bsAS* mutant wings. Also the interaction between *bsAS* TSS and middle4 region is lost in the mutant background. In **(C)** and **(D)**, error bars represent the SEM of at least three biological replicates. Statistical significance was computed by two-tailed t-test.

The only factor for which binding sites were found both at the TSS1 and *bsAS* TSS was the GAGA factor (GAF), for which no binding sites were found at TSS2. GAF ChIP-Seq data available for L3 wing imaginal discs [31] revealed strong GAF peaks at TSS1 and *bsAS* TSS (S6B Fig), providing additional support for a role of GAF in mediating the interaction. We also observed two strong Pol II peaks [32] at both TSSs in L3 wings, confirming that the two sites are transcriptionally active in this tissue at this developmental stage.

## Evolutionary conservation of *bs* and *bsAS*

We did not find any known amino acid motif in the 111-aa long region specific to the DSRF long isoform that could explain its specific function in the eye and in the vein regions of the wing. Actually, outside of the MADS box, which is strongly conserved across metazoans (and, intriguingly, systematically interrupted by an intron, in spite of dramatic changes in the exonic structure of the *bs* gene through evolution), there is little conservation of the DSRF protein, even already within diptera (Figs 5 and S7). By analyzing the existing gene annotation of the investigated species, which we have manually extended using RNA-Seq and computational evidence (see Material and Methods), we found, however, that the long-short isoform structure of *bs* gene appeared much earlier, at the root of pancrustacea (hexapoda and crustacea, Fig 5A).

Tracking the evolution of *bsAS* is far more complicated, since lncRNA generally show poor sequence conservation, even at close phylogenetic distances [64,65]. We have, therefore, used RNA-Seq data available across metazoans to characterize antisense transcription within the intron downstream the MADS box of *bs* gene. We have been able to trace the origin of *bsAS* also at the root of pancrustacea (Figs 5B and S7B), suggesting that both the long-short isoform structure of *bs* and *bsAS* emerged simultaneously. RNA-Seq data available from adult heads in *D. mojavensis* and *D. pseudobscura*, as well as from worker bees show that both *bs* and *bsAS* are less expressed in heads than in whole animals (Fig 5C and 5D), mirroring our findings in *D. melanogaster*. All these observations suggest that *bsAS* has appeared simultaneously to the long-short isoform structure of *bs*, possibly as a mechanism to regulate isoform usage.

## Discussion

*Blistered* encodes for DSRF (*Drosophila* Serum Response Factor), orthologous to the SRF protein in humans. In mammalian, SRF is a transcription factor involved in muscle and neural development [17,18]. In flies, *bs* has been studied both in the context of wing development and in memory establishment and persistence in adult flies [13–16]. In the fruit fly, two wing-specific enhancers are known to regulate *bs* expression [66], and it has been reported that *bs* expression is regulated in the wing by the EGF pathway, which negatively regulates its expression in the vein regions [14] and by HH pathway [66] and Vestigial and Scalloped proteins [67], which activate its expression in the wing pouch. However, to date, the regulation of the expression of the *bs* locus and how this regulation impacts *bs* function has not been fully elucidated. Here, we show that the expression of the NAT *bsAS*, one of the most highly expressed lncRNA in the fly wing, plays a crucial role in this regulation. The transcription of *bsAS* represses the expression of the long isoform, promoting, in consequence, the relative expression of the short isoform. Since the expression of *bsAS* seems also to prevent the activation of neural genes, we hypothesize that the expression of the long isoforms of *bs* may contribute to tissue neuralization, while the activation of the short isoform would prevent tissue neuralization.

More specifically, we have found that the *bs* locus is under the control of two different promoters: TSS1, driving the expression of both the short isoform B and the long isoform C, and interacting with the TSS of *bsAS*, and TSS2 driving the expression of the long isoform A only. Our results strongly suggest that TSS2 is active in the eye, where long isoform A is expressed.

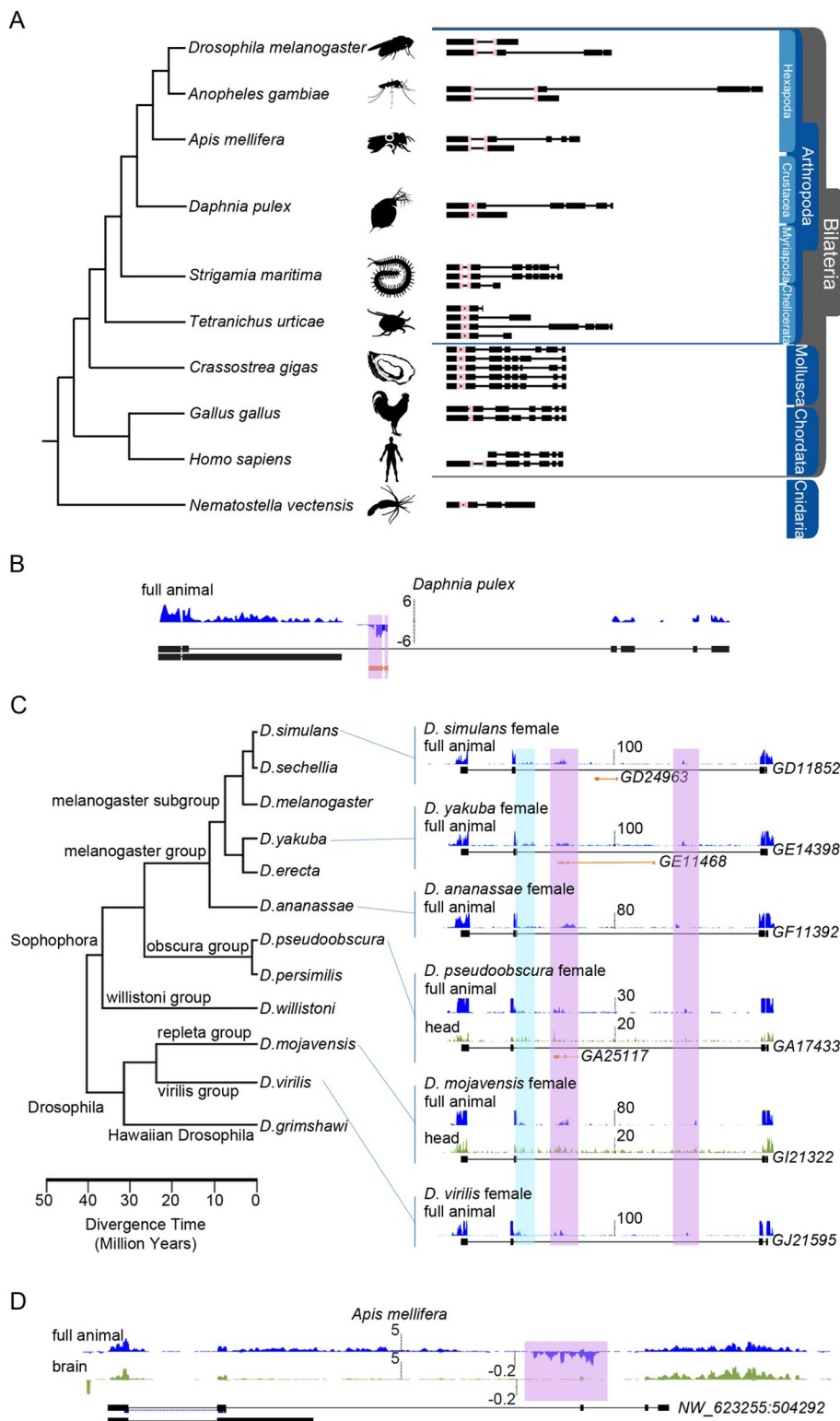

**Fig 5. Evolutionary conservation of *bs* and *bsAS*. (A)** Annotation of *bs* isoforms along metazoans. *bs* orthologous genes were manually annotated taking advantage of available RNA-Seq data in all species depicted. The pink box represents the MADS box, conserved along evolution. **(B)** Expression of *bs* and *bsAS* in *Daphnia pulex*. An antisense transcript is detected within the long intron of *bs* gene. **(C)** Expression of *bs* and *bsAS* in *Drosophila* species. Unstranded RNA-Seq tracks from modENCODE project are shown. When available, whole animals (in blue) and

heads (in green) RNA-Seq samples have been represented. The long isoform of *bs* is annotated in all species. We have been able to identify reads likely corresponding to the short isoform of *bs* (highlighted in blue). *D. simulans*, *D. yakuba* and *D. pseudoobscura* also present annotated antisense genes embedded into the long intron of the coding gene (in orange). In all species, reads corresponding to the *bsAS* gene have been identified (in purple). **(D)** Expression of *bs* and *bsAS* in *A. mellifera* whole animals (in blue) and brains (in green). As in fly, *bs* and *bsAS* are less expressed in brains compared to whole animals.

While TSS2 may also be active in the wing, our results suggest that TSS1 is the main driver of *bs* expression in this tissue, since long isoform C is much more abundant than isoform A. The wing is composed of both non-neural (interveins) and neural tissue (the nerves running along the veins [67]), and we have found the pattern of expression of the *bs* locus to be quite different in these two regions (S4 Fig). In interveins, the antisense *bsAS*, the promoter of which interacts with TSS1, is very highly expressed; this impairs the expression of the long isoforms (most likely C) and the short isoform B is predominantly expressed. Low expression of the long isoforms together with high expression of the short isoform, which could act as dominant negative further repressing the action of the long isoform, would lead to the differentiation of intervein tissue. In the vein regions, neither *bsAS* nor *bs* isoform B are expressed, and *bs* long isoform is likely to be the dominant one. We do not know, however, whether this is because TSS2 is the main promoter active in the vein regions, and therefore there is production only of the long isoform A, or because even though TSS1 is the main active promoter, there is specific downregulation of the *bsAS*, which cannot repress the synthesis of the long isoform C. Although the role of the long isoforms of *bs* during development is still unclear, several observations point to the idea that they may be related to neural development in the fly. First, we observed that *bsAS* mutants induced overexpression of *bs* long isoforms and of neural-specific genes in the wing (Fig 3C), promoting the appearance of extra-vein tissue. Second, *bs* has been reported to play essential roles in memory and learning in flies [15,16]. This fact altogether with the observation that *bs* long isoforms are the only forms of *bs* that are expressed in eye tissue (Fig 1C and 1D), suggest that these isoforms are the responsible for the neural functions assigned to *bs/DSRF*.

We would like to propose a model to explain how *bsAS* regulates the isoform usage of the *bs* gene in the wing to promote intervein differentiation (Fig 6). As mentioned above, in wing interveins, *bs* TSS1 and *bsAS* TSS are active, thus RNA Polymerases are recruited in both strands. TSS1 triggers the transcription of both the short isoform B and the long isoform C, but the elongation along both strands provokes the eventual collision of the polymerases and, in consequence, impairs the transcription of long isoform C. Thus, the short isoform B is the dominantly expressed in wing interveins. A collision model was previously suggested as responsible for sense-antisense gene regulation (for a review see [68]). The fact that the long isoforms are also detected in the interveins, albeit at low levels, may respond to a certain permissiveness of the collision. In contrast, when TSS2 is active (as in the eye) or *bsAS* is inactive, the RNA Polymerase is only recruited in the positive strand and the RNA elongation can occur unobstructed, resulting in the expression of the long isoforms of *bs*.

Nevertheless, it is difficult to hypothesize the mechanism by which the different DSRF protein isoforms impact cell fate, since we have not been able to find any known protein domain in the amino acid sequence specific to the long isoform. The only known domain in DSRF is a MADS box domain [13,69], also encoded in human SRF, and present in both isoforms.

The co-expression of *bs* isoform B and *bsAS* in the wing pouch prompted us to hypothesize a co-regulated expression of both genes. In this regard, we identified GAF as a potential mediator of both the contact and the co-regulation of *bs* TSS1 and *bsAS* TSS. GAF is a transcription factor known to form homodimers and also to interact with the RNA Polymerase machinery,

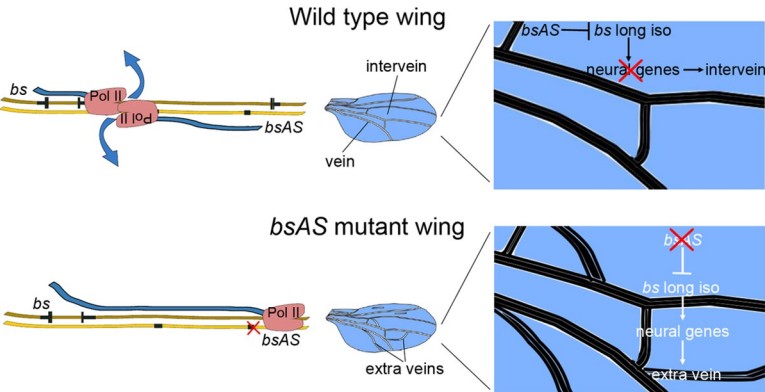

**Fig 6. Regulation of *bs* isoform by *bsAS*.** In the upper and bottom panels, positive and negative DNA strands are depicted as dark and light brown ribbons, respectively. Transcribed exons are depicted in black, narrow bars represent non-coding regions and wide bars represent coding regions. Blue ribbons represent nascent RNA. **Upper panel**. In wild type tissues, the transcription of *bsAS* inhibits the expression of *bs* long isoforms in the intervein regions, likely due to the collision of the RNA Polymerases elongating along both strands, which results in the short isoform being the predominant. **Lower panel**. In *bsAS* mutant wing intervein regions, as well as in eye tissues, the RNA Polymerase II is recruited only in the positive strand, allowing for the elongation until the end of the locus, and the long isoforms are fully transcribed. The expression of the *bs* long isoforms in intervein tissue induces the ectopic expression of neural genes and the appearance of extra veins in these regions.

insulator factors and chromatin remodelers [70]. Thus, we hypothesize that the interaction between *bs* TSS1 and *bsAS* TSS could be driven by GAF, either by dimerization of GAF proteins binding at the two TSSs or by its interaction with other factors. The deletion of *bsAS* TSS in *bsAS-/-* flies, that impairs the interaction between *bs* and *bsAS* promoters, includes the GAF binding site found at *bsAS* promoter region, indicating that this could be, indeed, the factor mediating the contact between both regions. GAF would then act as a transcriptional activator for both *bs* and *bsAS* in the intervein region of the wing. Indeed, the downregulation of the *Trithorax-like* (*Trl*) gene, encoding for GAF, induces a reduction of the intervein regions [71] and the appearance of extra vein tissue in adult wings [72], confirming a role of GAF in wing development. The loop formed by this interaction is likely stabilized by cohesin complexes, which have been demonstrated to contribute to interactions between chromatin regions such as enhancer-promoter communication [73].

Finally, our results suggest that *bsAS* has appeared simultaneously to the long-short isoform structure of *bs*, at the root of pancrustacea, a taxon that includes hexapods and crustaceans. Since non-insect hexapods and crustaceans are wingless animals, *bsAS* expression is ancestral to the development of the wings in insects. We hypothesize that, given that the neural *vs* non-neural expression pattern of *bsAS* is conserved at least within diptera and hymenoptera, *bsAS* regulation of isoform usage may be originally related to the differentiation of neural and non-neural structures in these animals. Alternatively, *bs* has been also related to the development of the tracheal system in the fly ([69,74]). Given that wings seem to have evolved from ancestral gills [75], which also have a respiratory function, one could hypothesize that the dual role of *bs* and its regulation by *bsAS* can also originate in the development of the tracheal system.

## Supporting information

**S1 Fig. *bsAS* is mainly expressed in *Drosophila melanogaster* wing tissues. (A)** Third instar larvae tissues used for RNA-Seq experiments. **(B)** Relationship between the correlation of exon ratio and antisense expression and the standard deviation (st. dev.) of sense gene exons ratio. Of the 145 SA pairs with both genes expressed in at least one tissue in L3, 104 pairs (72%,

102 genes) involve a protein coding gene with multiple isoforms. We computed the correlation between the expression of these NATs and the inclusion of the 964 exons of the sense protein coding genes. Because of the very small number of independent data points, we have little power to detect significant correlations. Still, we plotted this correlation against the standard deviation of the exon inclusion values, to focus specifically in the exons that changed the most (see Material and Methods). Among them, the case of *blistered -bs-* is the strongest, as it indeed shows strong positive and negative correlation between the expression of the NAT antisense to *bs* and the inclusion of three highly variable *bs* exons. **(C)** Ranking of expression of antisense lncRNAs in L3 (upper panel) and LP (lower panel) wings. *bsAS* is highlighted in red. **(D)** Histone methylation marks [77] at *bs* locus. There is a strong peak of H3K4me3 at *bs* TSS and *bsAS* TSS in WL3, whereas only a small peak at *bs* TSS is observed in EAL3. **(E)** H3K4me3 ChIP-qPCR in WL3. No differences are observed between *bs* TSS1 and TSS2 H3K4me3 marking. The deletion of *bsAS* TSS induces a dramatic reduction of H3K4me3 marking in *bsAS* TSS. TSS1 and TSS2 of *bs* do not show differences between *bsAS-/-* and wt. *RpL32* and *CG5367* correspond to positive and negative controls of the ChIP, respectively.
(TIF)

**S2 Fig. Depletion of the *bsAS* TSS region. (A)** *Drosophila* embryos expressing Cas9 nuclease in the germinal cells under the control of *vasa* driver, were injected with a mix of two plasmids expressing gRNAs against the TSS of *bsAS*. The screening was performed retroactively, by allowing putatively mutant flies to lie eggs before screening. Once the mutation was isolated, genetic crosses were performed to obtain the homozygous mutant flies. **(B)** PCR screening of *bsAS* deletion. A band of 800 bp was amplified in wt flies, whereas a band of 250 bp was obtained from homozygous *bsAS* mutant flies. **(C)** Alignment of wt and *bsAS-/-* genomic regions. A Single Nucleotide Polymorphism was detected in the *bsAS* TSS sequence (in green). The 5' region of the deletion has been cleanly repaired, but in the 3' region, several insertions/deletions have occurred (the insertion of 4 nucleotides–in blue-, the duplication of 9 nucleotides–in green- and the deletion of 2 nucleotides–in yellow-).
(TIF)

**S3 Fig. *bsAS* depletion induces strong phenotype in wings. (A)** Expression pattern of DSRF in *bsAS* mutant eye imaginal discs. Immunostaining of DSRF (red and grey) in wt (upper panels), *bsAS -/+* (middle panels) and *bsAS -/-* (lower panels) EL3. DSRF is overexpressed in a subset of cells posterior to the morphogenetic furrow in homozygous *bsAS* mutants. **(B)** Eyes of $w^{1118}$ (left panel) and *bsAS -/-* (right panel) adult males. No evident phenotype is observed in adult eyes, despite the overexpression of DSRF in third instar larvae. **(C)** Overexpression of *bsAS* under the control of *rotund* (*rn*) driver (specific driver that induces GAL4 expression in the wing pouch) in a *bsAS* mutant background, checked by qPCR. The overexpression of *bsAS* does not change the expression of any of *bs* isoforms. **(D)** DSRF staining on third instar larvae wings overexpressing the long isoform of *bs* under the control of *rn* driver. The overexpression of DSRF protein in L3 wings is significantly lower in wings overexpressing the long isoform of *bs* than in *bsAS* mutants. **(E-H)** Adult wings from males. **(E)** Wings overexpressing *bs* long isoform A under the control of the *nub* driver. They present extra vein tissue. **(F)** Wings overexpressing RNAi specific against the long isoform of *bs* in a *bsAS* homozygous mutant background. They show a partial rescue of the mutant phenotype, being less creased (compare to Fig 2G). **(G)** Wings expressing the short isoform of *bs* under the control the sal$^{E/Pv}$ driver. They present notches in the wing margin. **(H)** Wings expressing the short isoform of *bs* under the control the *rn* driver. The overexpression of the isoform B of *bs* induces high lethality and strong impairment of wing development.
(TIF)

**S4 Fig. Expression of *bsAS* and *bs* isoforms in vein and intervein regions. (A)** Co-localization of GFP and DSRF in *bs>GFP* heterozygous third instar larvae imaginal discs. **(B)** Expression of *bs* isoforms and *bsAS* in intervein (GFP positive) and vein (GFP negative) regions. *bsAS* and the *bs* short isoform B are more expressed in interveins than in veins, which are marked by high expression of the vein-specific genes *caupolican* (*caup*) and *rhomboid* (*rho*). In contrast, the long isoforms of *bs* are expressed at comparable levels in veins and interveins. Statistical significance was computed by one-sample t-test.
(TIF)

**S5 Fig. Interactions at the *bs* locus. (A)** RNA-Seq of Kc167 cells, from modENCODE [62]. *bsAS* is not expressed in these cells. *bs* expression is very low and isoform A seems to be the main expressed one. **(B)** Interaction of the *bs* locus and the neighboring regions (100 Kb). HiC data was obtained from Cubenas-Potts *et al.* [61]. The only strong interaction observed in this region is the contact between *bsAS* TSS and *bs* TSS1. No further interactions are observed.
(TIF)

**S6 Fig. Transcription Factor binding at the *bs* locus. (A)** Profile of DNA binding of 24 transcription factors during embryogenesis from modENCODE project [63]. Middle4 region is very close to a hot spot for transcription factor binding. **(B)** Profile of DNA binding of GAF [31] (light green) and RNA Pol II [32] (dark green) in third instar larvae wings. GAF and RNA Pol II peaks coincide with the expression of *bs* TSS1 and *bsAS* in wings. Blue vertical lines represent GAF binding sites at *bs* TSS1 and *bsAS* TSS.
(TIF)

**S7 Fig. Evolutionary conservation of *bs* and *bsAS*. (A)** Multiple sequence alignment of the long *bs* isoform using MAFFT and visualization of the frequency-based difference using NCBI MSA Viewer. High sequence conservation is observed between 160–270 bp where MADS-box is located. Sequence conservation drops rapidly outside this region. **(B)** Read-depth along *bs* locus. Organisms are sorted by the tree of life. Number of split reads are highlighted in the exon junctions generated using ggsashimi [39]. Stranded RNA-Seq is shown in two separated strands, where the negative strand is negated and shown below the positive strand. When available, whole animals (in blue) and heads (in green) RNA-Seq samples have been represented. The long isoform of *bs* is annotated in all species. Expression of both long and short isoforms of *bs* is supported by read coverage and by split alignments in exon junctions in represented species. Antisense expression is supported by stranded read coverage and by split alignments in the exon boundaries in the Diptera *Drosophila melanogaster*, in the Coleoptera *Anoplophora glabripennis* and in the Crustacea *Daphnia pulex*. Stranded RNA-Seq of the Hymenoptera *Apis mellifera* presents more antisense signal in the whole body compared to the head sample, however there are no split reads supporting exon junctions. The wingless basal Hexapoda, *Folsomia candida*, presents a clear set of split reads supporting the junctions of *bsAS* and no split reads shared between *bs* and *bsAS*, however the data is unstranded. Overall, Hexapoda and Crustacea show expression of both long-short *bs* isoforms as well as the antisense expression on the lncRNA *bsAS*.
(TIF)

**S1 Table. Summary statistics of RNA-Seq samples.** All samples generated along the manuscript are summarized here.
(PDF)

**S2 Table. Gene and transcript expression of *bs* and *bsAS* in wt and *bsAS*-/- tissues.** WL3, third instar larvae wing. WLP, late pupa wing. EAL3, third instar larvae eye-antenna. ELP, late

pupa eye. Genes are highlighted in grey. Expression values are represented in TPMs.
(PDF)

**S3 Table. DEG between wt and *bsAS*-/- in L3 and LP wings.** Expression values are represented in TPMs. L3 DEG are represented in sheet 1 and LP DEG are represented in sheet 2.
(XLSX)

**S4 Table. List of primers used to perform the study.** Name of the primer, sequence and experiment in which it has been used are depicted.
(XLSX)

## Acknowledgments

We thank Bruna R. Correa, Ramil Nurtdinov, Sebastian Ullrich and Beatrice Borsari for helpful discussions about the data and the manuscript and Romina Garrido for administrative assistance. We also thank the Ultrasequencing, the Flow Cytometry and the Advanced Light Microscopy Units of the CRG (Barcelona, Spain), for sample processing and the Rainbow Transgenic Flies Inc. (Camarillo, USA) and the *Drosophila* Injection Service from the Institut de Recerca Biomèdica (IRB) (Barcelona, Spain) for injection services. We also thank Montserrat Corominas and Florenci Serras, from the University of Barcelona, for the anti-DSRF antibody and for *Drosophila* strains and Fátima Gebauer, from the Center for Genomic Regulation, for *Drosophila* strains. This research reflects only the authors' views and the Community is not liable for any use that may be made of the information contained therein.

## Author Contributions

**Conceptualization:** Sílvia Pérez-Lluch, Cecilia C. Klein, Alessandra Breschi, Marina Ruiz-Romero, Roderic Guigó.

**Formal analysis:** Sílvia Pérez-Lluch, Cecilia C. Klein, Alessandra Breschi.

**Funding acquisition:** Roderic Guigó.

**Investigation:** Sílvia Pérez-Lluch, Cecilia C. Klein, Alessandra Breschi, Marina Ruiz-Romero, Amaya Abad, Lyazzat Bekish, Carme Arnan.

**Project administration:** Roderic Guigó.

**Resources:** Roderic Guigó.

**Software:** Emilio Palumbo.

**Supervision:** Sílvia Pérez-Lluch, Roderic Guigó.

**Validation:** Sílvia Pérez-Lluch, Amaya Abad.

**Visualization:** Sílvia Pérez-Lluch, Cecilia C. Klein, Alessandra Breschi, Marina Ruiz-Romero.

**Writing – original draft:** Sílvia Pérez-Lluch, Cecilia C. Klein, Alessandra Breschi, Marina Ruiz-Romero, Roderic Guigó.

**Writing – review & editing:** Sílvia Pérez-Lluch, Roderic Guigó.

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
