## [Decision Letter · Decision Letter 0]

10 Mar 2020

Dear Dr Guigo,

Thank you very much for submitting your Research Article entitled 'bsAS, an antisense long non-coding RNA, essential for correct wing development through regulation of blistered/DSRF isoform usage' to PLOS Genetics. Your manuscript was fully evaluated at the editorial level and by independent peer reviewers. The reviewers appreciated the attention to an important problem as well as the novelty of the findings. However, both reviewers raised substantial concerns about the current manuscript. Based on the reviews, we will not be able to accept this version of the manuscript, but we would be willing to review a much-revised version. We cannot, of course, promise publication at that time.

If you decide to revise the manuscript for further consideration at PLOS Genetics, please aim to resubmit within the next 60 days, unless it will take extra time to address the concerns of the reviewers, in which case we would appreciate an expected resubmission date by email to plosgenetics@plos.org.

[LINK]

We are sorry that we cannot be more positive about your manuscript at this stage. Please do not hesitate to contact us if you have any concerns or questions.

Yours sincerely,

Claude Desplan

Associate Editor

PLOS Genetics

Gregory P. Copenhaver

Editor-in-Chief

PLOS Genetics

Reviewer's Responses to Questions

**Comments to the Authors:**

Reviewer #1: In this work, Lluch and colabs searched for Natural Antisense Transcripts that could have a function during development by strand-specific RNA seq in different tissues of the fly. The authors identified bsAS as an antisense lncRNA to the bs/DSRF gene. bs is requiered for the proper vein-intervein pattern in the wing, trachea formation and also is implicated in neural processes such as memory, vision and sleep. Two different issoforms are described for bs. A long isoform (encoded by the A and C transcripts) and a short one (B transcript), being the later the one expressed in the wing disc. Interestingly, the authors beautifully demostrated that bsAS is expressed in the same intervein pattern as bs in the wing and that bsAS is requiered in cis to determine wich isoform is expressed. Mutants for bsAS showed a strong derepression of the long isoform without affecting the expression of the short one. Comparing the wing phenotypes of the bsAs mutants with the ectopic expression of the long isoform, the authors proposed that the intervein or neural cell fates are specified by the different ussage of the bs isoforms. Moreover, using 3C assays demostrated a physical contact between the transcription start site of bsAs and of the short isoform, suggesting a co-regulation. In addition, the authors perform a evolutionary analysis to propose a common origin of the bsAS and the long isoform structure of the bs gene.

The paper is well organized, written and the conclussions are novel and mostly well supported by their experiments. However, I have some discrepancies with some of the conclusions drawn from their results and few comments and questions that the authors may consider before publication.

1) One of my main problems with their model is that the authors conclude that the expression of the long isoform induces tissue neuralization (for example Fig. 5). In the text the authors mistake the vein as a neural tissue misleading the reader in several paragraphs in the text (pg. 17 for example). Also, the long isoform is normally not expressed in the wing and therefore, unless it is proven, is not requiered for vein patterning. Is it reasonable to think, as suggested by the authors, that the different functions of bs are carried out by the different isoforms, however this hypothesis was not tested in the paper.

A simple way to interpret their results is that the expression of the long isoform in the bsAS mutants acts as a dominant negative version of the short isoform, generating a partial bs loss of function phenotype and the formation of extraveins. Maybe looking at vein patterning genes in the different mutant combinations could help understand the phenotypes and function of the different isoforms. Also, I think is important to show the expression of bs, of the two different isoforms and of bsAS in the different tissues such as wing vs eye –disc.

2) In the discussion the authors “hypothesize that bs short isoform can counteract the expression of the long isoforms in the intervein regions, silencing the expression of neural genes and avoiding the development of ectopic veins within intervein regions”. However, if I understood correctly the paper and their results, is not the short isoform, but the transcription of the bsAS lncRNA the responsible for the downregulation of the long isoform in the intervein region. Again, the role of the long isoform inducing neural genes and ectopic vein development is not convincingly demosntrated.

It is a little confussing to me the different uses of isoform A or C in figure 5. They have shown that neither isoform are expressed in the wild type wing and only in the bsAS mutants, the isoform C is derepressed (pg. 14). So why picture iso A in the vein territories in fig. 5?

3) The upregulation of bs expression in the bsAS mutant is not evident at the RNA level in Fig. 2 and at the protein level it is hard to convince the readers comparing two different discs without any quantification. If the authors are convinced that there is an upregulation of Bs protein, maybe a WB could be more informative and quantitative.

Also, it is possible that the upregulation of the total bs levels is not detected because the derepression of the long isoform is very low compared to the short one, and therefore even an upregulation of the long one maybe masked and difficult to detect. I think this has to be clarified better.

4) In a bs mutant the entire intervein domain is transformed into vein tissue (see Fig. 2C Roch et al, 1998). This phenotype is different from what is observed in the homozygoes mutant for bsAS where some extra vein tissue is observed but most of the intervein region is preserved. I don't think the phenotypes are equivalent because in the bsAS mutant the short isoform is still normally expressed. The authors conclude that the bsAS mutant phenotype is a consequence of the overexpression of the long isoform. To demonstrate this important conclusion the authors could use specific RNAis available to the long isoforms such as JF02319 and KK108659 in the bsAS mutant background to test if the bsAS mutant phenotype could be restore.

5) Do the authors find any vein promoting gene upregulated in their RNA seq of bsAS mutant wings?. How is the pattern of genes such Knirps (kni) and Iroquois (Iro) in bsAS mutants?.

6) The authors showed a co-regulation of bsAS and the TSS1 using 3C techniques. As both bs and bsAS are expressed in the same intervein pattern, are they regulated by the same cis-regulatory module?. Is there a physical interaction between those regions with the identified wing CRM of bs (Nussbaumer et al, 2000) ?

Minor comments:

-p Value in Figure 1D and 2B for CG4812.

-is bs expressed in the EA disc? Maybe including a picture could be informative.

-The comparation and p values are not clear in Fig. 3C and D.

-Wl3 track for bs expression in Fig. Sup2I. The expression of the bs isforms is not observed. Compare it with Fig. 1C. Why?

Reviewer #2: In this study, Perez-Lluch et al., investigate the role of an antisense long non-coding RNA (lncRNA) in regulating usage of the bs/DSRF transcripts and the subsequent role in wing development. First, they describe the characterization of lncRNAs. First they identify the bs/DSRF gene as a potential candidate for transcript usage depending on lncRNA. Comparison of bs/DSRF transcript usage reveal strong differences in wing versus leg or eye-antennal discs. bs/DSRF has been shown to be expressed in the presumptive intervein tissue of the wing disc and the antisense transcript (bsAS) appears to be expressed in a similar pattern. Using a CRISPR/CAS9 strategy they make a specific deletion of the bsAS. This deletion results in the strong increase in the expression of the bs/DSRF long transcripts and in severe wing phenotype. The authors interpret this phenotype as a consequence of high bs/DSRF long transcript; however when over-expressing the bs/DSRF long transcript they observe minor (Fig 2H) or no visible phenotype (Fig S2E-F). Next, they show direct interactions between the transcription start site of bs and bsAS. Finally, the author analyze throughout evolution the conservation of the dual transcripts of the bs/DSRF orthologues. Experiments are properly performed to address the issue of the bsAS function. However there a major problem with the interpretation. The bsAS deficiency induces an extreme phenotype that is not at all comparable to the one induced by over-expression of the bs/DSRF long transcript. Thus, they cannot postulate that the extreme wing phenotype is due to the increased expression of the bs/DSRF long isoform. The conclusion (p14 bottom) and the model (Fig5) are over-estimated. I doubt that this study can be published in PLoS Genetics, unless they provide a consistent explanation based on experimental data for the severe wing phenotype observed in bsAS deficiency.

Additional comments:

1. Ref13 (Montagne et al., 1996) should be mentioned as the same time as ref10 (Fristrom et al., 1994) since ref13 is the one showing that bs and DSRF are allelic.

2. References for fly strains in M&M must be provided.

3. In situ hybridization in wing disc for bsAS and bs are not the “same’, they exhibit an overlapping pattern. The text should be modified.

4. The author cannot compare Fig2C (immunostaining to bs/DSRF) with Fig1E (in situ hybridization). Actually, there is no control for the genuine protein level in wild type disc in Fig2C.

5. bs/DSRF protein levels should be compared in the same experiment for wild type, for bsAS deficient and for long transcript-expressing wing discs.

6. The phenotype of bsAS heterozygous deficiency is minor, nonetheless, resembles the one of bs/DSRF long transcript over-expression; the author should also compare bs/DSRF protein levels in both genetic context.

**Have all data underlying the figures and results presented in the manuscript been provided?**

Reviewer #1: Yes

Reviewer #2: Yes

PLOS authors have the option to publish the peer review history of their article (what does this mean?). If published, this will include your full peer review and any attached files.

Reviewer #1: No

Reviewer #2: No

---

## [Decision Letter · Decision Letter 1]

29 Sep 2020

Dear Dr Guigo,

Thank you very much for submitting your Research Article entitled 'bsAS, an antisense long non-coding RNA, essential for correct wing development through regulation of blistered/DSRF isoform usage' to PLOS Genetics. Your manuscript was evaluated by the two initial independent peer reviewers. The reviewers found the paper much improved but still asked for you to address a few remaining points before the paper can be accepted. This should mostly require editorial changes

We therefore ask you to modify the manuscript according to the review recommendations before we can consider your manuscript for acceptance. Your revisions should address the specific points made by each reviewer.

[LINK]

Yours sincerely,

Claude

Claude Desplan

Associate Editor

PLOS Genetics

Gregory P. Copenhaver

Editor-in-Chief

PLOS Genetics

Reviewer's Responses to Questions

**Comments to the Authors:**

Reviewer #1: I acknowledge the effort and the experiments done by the authors to respond to the reviewers comments and suggestions. Overall, I really like the paper and it deserves to be published in PloS Genetics. The experiments proven the role of the LncRNA bsAS as a regulator of bs isoform usage are very solid.

However, I still identify some problems with the interpretation of the data that were raised in the first revision. Specifically, I find the the role of the bs long isoform as a determinant of vein fate during normal development not been proved. Moreover, the authors insists in consider the vein tissue as a neural tissue and in my opinion this is an inaccuracy and could lead to confussion.

To prove the role of the bs long isoform as a determinant of vein tissue in the wing, the authors should demonstrate:

-1) bs long isform expression in the vein region. In situ hybridization of the bs long isoform?

-2) the funtion of the bs long isform in a wt background in the wing. A simple experiment would be to knockdown the bs long in the wing with an specific RNAi. If their model (fig. 6) is correct, the prediction would be wings with defective veins. However, in fig. 2J (although no ideal as it is performed in the bs+/- background) no defect on the characteristic vein pattern is observed.

Sentences like “In contrast, in the vein regions, in which bsAS is poorly expressed, the long isoforms (A or C, or both) are the dominant ones, inducing the expression of neural genes and the differentiation of veins” and their model in fig. 6 lead to a function of the long isform as determinant of vein fate that has not been proven.

The authors also include a new experiment where cells GFP+ and GFP- cells are isolated of third instar larvae wings of bs-GAL4>UAS-GFP flies and conclude that the bsAS and short isoforms are expressed mainly in intervein (GFP+) while long isoforms are expressed at similar levels in vein and intervein. GFP- cells are formed by vein cells (minority) and hinge and notum cells (mayority), and therefore this experiment is not very precise, informative and conclusive.

Reviewer #2: The revised version of the manuscript of Perez-Lluch et al., is more convincing than the first one. My main concerns have been addressed; in particular, immunostaining pictures that strengthen the relationship between bsAS activity and bs protein level expression, and novel data showing that selective knockdown of the bs/DSFR long isoform can significantly suppress the wing phenotype of bsAS-/- mutants.

Minor point: In the discussion, the authors stress the ancestral conservation of bs and bsAS throughout evolution, even in wingless animals, and conclude by an original function in neural differentiation. However, wings have been proposed to have evolved from ancestral gills (doi.org/10.1038/385627a0). Further, dSRF plays a critical role in the differentiation of the terminal tracheal cells (Guillemin et al 1996 Development 122:1353-62). Therefore, one can also speculates that the ancestral conservation of bs and bsAS may rely on an original function in the development of the respiratory system. Although it would be easy to look at the differentiation of the terminal tracheal cells in bsAS mutant flies, I consider that it is not the purpose of the present study. Nonetheless, I suggest a modulation of the neural differentiation origin in the conclusion.

**Have all data underlying the figures and results presented in the manuscript been provided?**

Reviewer #1: Yes

Reviewer #2: Yes

PLOS authors have the option to publish the peer review history of their article (what does this mean?). If published, this will include your full peer review and any attached files.

Reviewer #1: No

Reviewer #2: No

---

## [Editor Report · Decision Letter 2]

3 Nov 2020

Dear Dr Guigo,

We are pleased to inform you that your manuscript entitled "bsAS, an antisense long non-coding RNA, essential for correct wing development through regulation of blistered/DSRF isoform usage" has been editorially accepted for publication in PLOS Genetics. Congratulations!

Yours sincerely,

Claude Desplan

Associate Editor

PLOS Genetics

Gregory P. Copenhaver

Editor-in-Chief

PLOS Genetics

Comments from the reviewers (if applicable):

**Data Deposition**

http://datadryad.org/submit?journalID=pgenetics&manu=PGENETICS-D-20-00125R2

**Press Queries**

---

## [Editor Report · Acceptance letter]

4 Dec 2020

PGENETICS-D-20-00125R2 

*bsAS*, an antisense long non-coding RNA, essential for correct wing development through regulation of *blistered/DSRF* isoform usage 

Dear Dr Guigó, 

We are pleased to inform you that your manuscript entitled "*bsAS*, an antisense long non-coding RNA, essential for correct wing development through regulation of *blistered/DSRF* isoform usage" has been formally accepted for publication in PLOS Genetics! Your manuscript is now with our production department and you will be notified of the publication date in due course.

With kind regards,

Livia Horvath

PLOS Genetics

On behalf of:
